# A secondary *RET* mutation in the activation loop conferring resistance to vandetanib

Takashi Nakaoku[1], Takashi Kohno[1,2], Mitsugu Araki[3,4], Seiji Niho[5], Rakhee Chauhan[6], Phillip P. Knowles[6], Katsuya Tsuchihara[2], Shingo Matsumoto[2,5], Yoko Shimada[1], Sachiyo Mimaki[2], Genichiro Ishii[7], Hitoshi Ichikawa[2], Satoru Nagatoishi[8], Kouhei Tsumoto[8], Yasushi Okuno[4], Kiyotaka Yoh[5], Neil Q. McDonald[6,9] & Koichi Goto[5]

Resistance to vandetanib, a type I RET kinase inhibitor, developed in a patient with metastatic lung adenocarcinoma harboring a *CCDC6-RET* fusion that initially exhibited a response to treatment. The resistant tumor acquired a secondary mutation resulting in a serine-to-phenylalanine substitution at codon 904 in the activation loop of the RET kinase domain. The S904F mutation confers resistance to vandetanib by increasing the ATP affinity and autophosphorylation activity of RET kinase. A reduced interaction with the drug is also observed in vitro for the S904F mutant by thermal shift assay. A crystal structure of the S904F mutant reveals a small hydrophobic core around F904 likely to enhance basal kinase activity by stabilizing an active conformer. Our findings indicate that missense mutations in the activation loop of the kinase domain are able to increase kinase activity and confer drug resistance through allosteric effects.

[1] Division of Genome Biology, National Cancer Center Research Institute, 5-1-1, Tsukiji, Chuo-ku, Tokyo 1040045, Japan. [2] Division of Translational Genomics, Exploratory Oncology Research and Clinical Trial Center, National Cancer Center, 5-1-1, Tsukiji, Chuo-ku, Tokyo 1040045, Japan. [3] Advanced Institute for Computational Science, RIKEN, 7-1-26 Minatojima-minami-machi, Chuo-ku, Kobe-city, Hyogo 6500047, Japan. [4] Department of Clinical System Onco-Informatics, Graduate School of Medicine, Kyoto University, 54 Kawaracho, Shogoin, Kyoto-city, Kyoto 6068507, Japan. [5] Department of Thoracic Oncology, National Cancer Center Hospital East, 6-5-1, Kashiwanoha, Kashiwa-city, Chiba 2778577, Japan. [6] Signaling and Structural Biology Laboratory, The Francis Crick Institute, London NW1 1AT, UK. [7] Division of Pathology, Exploratory Oncology Research and Clinical Trial Center, National Cancer Center, 6-5-1, Kashiwanoha, Kashiwa-City, Chiba 2778577, Japan. [8] Medical Proteomics Laboratory, Institute of Medical Science, The University of Tokyo, 4-6-1, Shiroganedai, Minato-ku, Tokyo 1088639, Japan. [9] Institute of Structural and Molecular Biology, Department of Biological Sciences, Birkbeck College, Malet Street, London WC1E 7HX, UK. Correspondence and requests for materials should be addressed to T.K. (email: tkkohno@ncc.go.jp)

Oncogenic *ALK* and *ROS1* fusion-targeted therapy using type I tyrosine-kinase inhibitors (TKIs), which bind to the ATP-binding cleft of kinases, is highly effective in lung adenocarcinoma (LADC)[1,2]; however, such cancers inevitably acquire resistance to targeted therapies, which severely limits the efficacy of cancer treatments. Secondary mutations that cause amino acid substitutions in the kinase domain (KD), including the gatekeeper and solvent-accessible regions, are an important cause of resistance to various extents[3]. The identification of resistance mutations in ALK and ROS1 led to the development of novel TKIs to overcome acquired resistance[1,3,4].

Oncogenic fusions of the *RET* kinase gene are present in 1–2% of LADCs[5,6], and are the subject of intense investigation. These fusions are promising targets for the treatment of LADC[7,8], because of the availability of clinically active RET TKIs, such as vandetanib and cabozantinib[9]. However, the mechanisms underlying acquired resistance to RET TKIs in lung cancer patients remain to be elucidated, and the molecular process by which cancer cells acquire such resistance needs to be investigated. Here we report the first case of a secondary *RET* mutation associated with resistance to the RET TKI vandetanib. The patient described was enrolled into our clinical trial[8], LURET (Lung Cancer with RET Rearrangement Study; clinical trial registration number: UMIN000010095, https://upload.umin.ac.jp/), which investigates the efficacy of vandetanib for the treatment of non-small cell lung cancer (NSCLC) with oncogenic *RET* fusion. In this trial, 19 RET fusion-positive cases were enrolled through genetic screening of 1536 patients, and 17 eligible cases showed a response rate of 53% and a progression-free survival period of 4–7 months[8].

## Results

**Case report**. A 57-year-old Japanese woman was referred to our hospital with a nodule in her left lung that was detected in a medical checkup. Bronchoscopic and mediastinoscopic examinations revealed adenocarcinoma of the lung with mediastinal lymph node metastases. The patient underwent concurrent chemoradiotherapy with cisplatin and vinorelbine, resulting in a partial response; however, 2 years later, multiple bone metastases developed. Genetic examination revealed no mutation in *EGFR*. The patient received second- to sixth-line chemotherapies consisting of gefitinib, pemetrexed, docetaxel, gemcitabine, and S-1. During sixth-line chemotherapy, the patient developed right cervical lymphadenopathy (Fig. 1a and Supplementary Fig. 1), and a biopsy of the lymph node revealed adenocarcinoma (Fig. 1b and Supplementary Fig. 2a). Additional molecular testing for *RET*, *ALK*, and *ROS1* fusions was performed by LC-SCRUM (Lung Cancer Genomic Screening Project for Individualized Medicine in Japan)[10]. Reverse transcriptase-polymerase chain reaction (RT-PCR) analysis of total RNA extracted from snap-frozen biopsied tumor cells revealed a *CCDC6-RET* fusion and no other fusions (Fig. 1c). The *CCDC6-RET* fusion led to the expression of a fusion transcript in which exon 1 of *CCCDC6* was joined to exon 12 of *RET*. The *CCDC6-RET* fusion was validated by identifying breakpoint junctions in genomic DNA

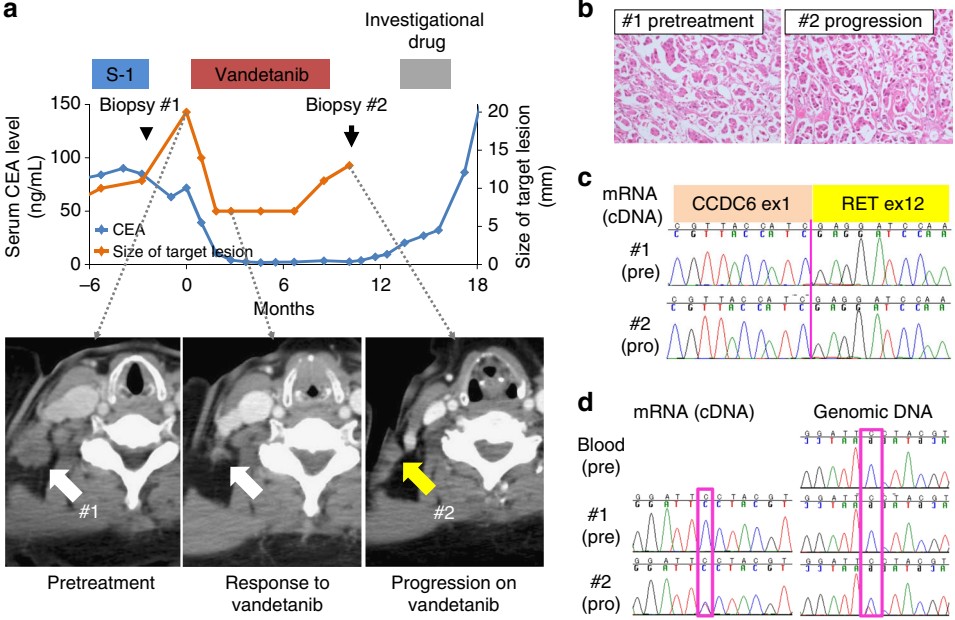

**Fig. 1** Identification of a RET-S904F mutation conferring resistance to vandetanib. **a** Clinical course of the patient and axial chest computed tomographic (CT) scan. (Upper) The blue line indicates the serum CEA level, and the orange line indicates the size of the target lesion (the right metastatic cervical lymph node). The time points of the biopsy of metastatic lymph nodes are indicated by an arrowhead in Biopsy #1 and an arrow in Biopsy #2 (the details of the clinical course are shown in Supplementary Fig. 1). (Lower) CT scan images of the metastatic lymph node as a target lesion. **b** Sanger sequencing results of RT-PCR products from pretreatment specimens (Biopsy #1, pre) and specimens obtained at disease progression (Biopsy #2, pro). The same *CCDC6-RET* fusion transcript in which exon 1 of *CCCDC6* is joined to exon 15 of *RET* was expressed. **c** Histological findings of hematoxylin/eosin-stained lymph node biopsy specimens obtained before treatment (Biopsy #1) and after disease progression (Biopsy #2). The identical pathological features are shown. **d** Sanger sequencing of genomic-PCR and RT-PCR products from peripheral blood, pretreatment specimens (pre), and specimens obtained at disease progression (pro). A mutation of cytosine to thymine at residue 2902 was detected only in the resistant tumor specimen. Genomic and RT-PCR analysis was performed using a primer in *CCDC6*-exon 1 and a reverse primer in RET-exon 15. The detection of the substitution, which causes an amino acid substitution of serine-to phenylalanine at codon 904 (in magenta), in genomic DNA and in the fusion transcript suggested that the mutation occurred on the rearranged *RET* allele in the resistant tumor

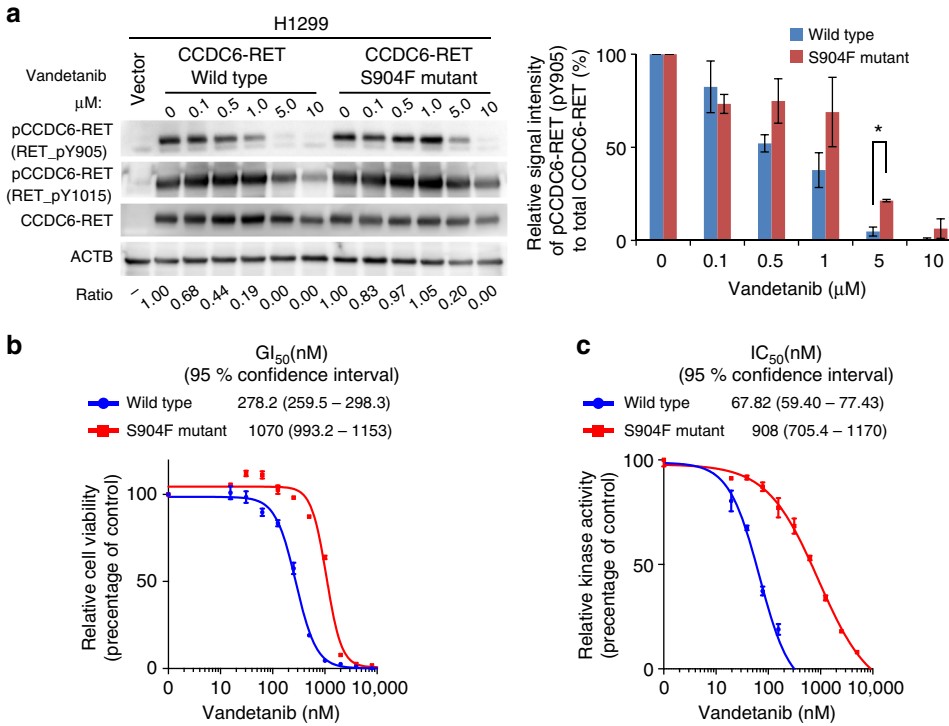

**Fig. 2** Resistance to vandetanib by RET-S904F mutation. **a** Immunoblot analysis of the wild type and S904F mutant CCDC6-RET fusion proteins. (Left) An expression vector encoding full-length wild type or S904F mutant *CCDC6-RET* cDNA was transiently introduced into H1299 lung cancer cells. After exposure to the indicated concentrations of vandetanib for 6 h, phosphorylation at tyrosines 905 and 1015 were detected. The signal intensities, calculated as the ratio of phosphorylated to total CCDC6-RET fusion proteins, are shown at the bottom. (Right) The experiment (shown on the left) was performed separately three times (Supplementary Fig. 8). The graph shows the mean ratios of phospho-Ret (pTyr905) to total RET from three separate experiments with standard deviations (shown as error bars). Concentrations showing statistical significance ($p < 0.05$ by $t$-test) are marked with an asterisk. ACTB: beta-actin. **b** Cell growth assay using IL3-independent Ba/F3 cells carrying lentivirally transduced *CCDC6-RET* cDNA with or without the S904F mutation. Ba/F3 cells (2000 cells) were plated in quadruplicate in 96-well plates and treated with serially diluted vandetanib. After incubation for 72 h, cell viability was measured using the CellTiter-Glo luminescent reagent with EnVision and the viability curves with standard deviations (shown as error bars) were calculated with GraphPad Prism version 6.0. Reproducibility was confirmed by performing the same experiment three times. **c** In vitro kinase assays. Recombinant RET kinase domain (KD) (amino acids 658–1072) with or without the S904F mutation was expressed by baculovirus in Sf9 insect cells using an N-terminal GST tag (gene accession number: NM_020630). The assay was initiated by addition of $^{32}$P-ATP to the reaction mixture and serially diluted vandetanib, and the IGF1Rtide synthetic peptide was added as the substrate (KKKSPGEYVNIEFG). After incubation at 30 °C for 20 min, the radioactivity was measured in a TriLux scintillation counter after transferring the products onto a phosphocellulose P81 plate. IC50 values with 95% confidence interval were calculated using GraphPad Prism version 6.0. Error bars show standard deviations. Reproducibility was confirmed by performing the same experiment three times

(Supplementary Fig. 2b). The patient was subsequently enrolled into the LURET trial.

The patient showed a dramatic response to vandetanib, a type I RET TKI, with reduction in her tumor size from 20 to 7 mm in diameter at 12 weeks. This was consistent with a high-response rate in the LURET study in *CCDC6-RET*–positive cases (5/6 cases, 83%)[8]. However, the patient developed a resistant tumor with the same histological features and a *CCDC6-RET* fusion (C1; R12) at 38 weeks (Fig. 1a–c, Supplementary Fig. 1 and Fig. 2a). Given the high diversity of breakpoints for *RET* fusions[11], the identical genome structures of the breakpoint junctions (Supplementary Fig. 2b) indicated that the resistant tumor originated from the original tumor present before vandetanib treatment.

**Discovery of a S904F secondary mutation in the RET kinase domain.** Targeted deep sequencing of cancer-related genes from genomic DNA identified a serine-to-phenylalanine substitution at codon 904 (S904F) in RET as the only non-synonymous mutation in the resistant tumor; the baseline tumor samples contained no non-synonymous mutations (Supplementary Fig. 3). Sanger

sequencing of RT-PCR products of fusion transcripts verified that the S904F mutation occurred on the *RET* allele fused to the *CCDC6* locus (Fig. 1d). Whole-exome sequencing analysis revealed four mutations specific to the resistant tumor, including the *RET*-S904F mutation (Supplementary Fig. 3b). The lack of reports connecting the three other mutated genes to drug resistance prompted us to further study the *RET*-S904F mutation.

**Vandetanib resistance is related to increased ATP affinity and autophosphorylation activity.** The serine 904 residue is located in between two autophosphorylation tyrosines Y900 and Y905 within the canonical activation loop (AL) of RET kinase; however, it is not conserved among kinases (Supplementary Fig. 4). Phosphorylation of an exogenously expressed S904F mutant of CCDC6-RET in H1299 lung cancer cells, which do not express endogenous RET[6], was maintained at a higher vandetanib concentration (half-maximal inhibitory concentration [IC50] = 2.6 μM) than that in the wild-type CCDC6-RET protein (IC50 = 0.57 μM) (Fig. 2a). This finding was validated in another cell line, Ba/F3, which grew in an interleukin 3 (IL3)-independent manner

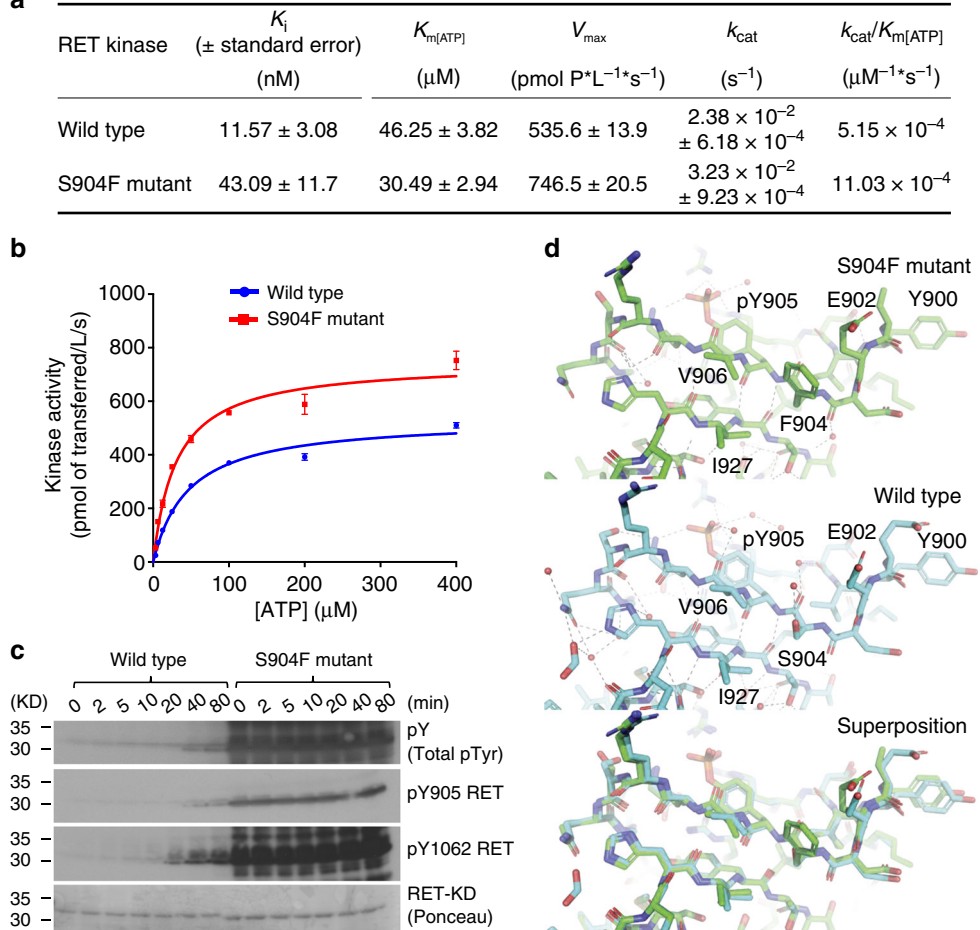

**a**

| RET kinase | $K_i$ (± standard error) (nM) | $K_{m[ATP]}$ (μM) | $V_{max}$ (pmol P*L$^{-1}$*s$^{-1}$) | $k_{cat}$ (s$^{-1}$) | $k_{cat}/K_{m[ATP]}$ (μM$^{-1}$*s$^{-1}$) |
|---|---|---|---|---|---|
| Wild type | 11.57 ± 3.08 | 46.25 ± 3.82 | 535.6 ± 13.9 | $2.38 \times 10^{-2}$ ± $6.18 \times 10^{-4}$ | $5.15 \times 10^{-4}$ |
| S904F mutant | 43.09 ± 11.7 | 30.49 ± 2.94 | 746.5 ± 20.5 | $3.23 \times 10^{-2}$ ± $9.23 \times 10^{-4}$ | $11.03 \times 10^{-4}$ |

**Fig. 3** Kinetic and structural properties of the RET-S904F mutant. **a** Enzyme kinetics parameters of wild type and S904F mutant RET KD proteins. The kinase assay was performed in triplicate at 25 °C for different incubation times (0, 10, 20, 30, 40, and 50 min) using purified RET KDs, increasing concentrations of ATP (3.125–400 nM), and serially diluted vandetanib. The assay was initiated by addition of $^{32}$P-ATP, and the reaction mixture was incubated at 30 °C. These experiments were performed independently three times. The data were analyzed using GraphPad Prism version 6.0 to calculate kinetic parameters, $K_i$, and IC$_{50}$ values. **b** Saturation curve graphed by Michaelis–Menten equation. The graph with standard deviations (shown as error bars) was generated using GraphPad Prism. **c** Western blotting showing the autophosphorylation time course in the wild type and S904F mutant RET KDs. Phosphorylation of the recombinant purified RET KDs treated with ATP (5 mM) and MgCl$_2$ (10 mM) for 0–80 min was detected with the indicated antibodies. **d** Comparison of S904F and wild-type RET structures. (Upper) Detail of side chain contacts close to the F904 mutation site of mutant RET kinase domain (PDB code 6FEK). Selected sidechains are labeled. Bound waters are shown as red spheres and hydrogen-bonds drawn in gray as defined by Pymol Molecular Graphics System (Schrödinger, LLC, New York, NY). (Middle) Detail of side chain contacts close to the S904 of wild-type RET kinase domain (PDB code 2IVT), colored as per panel (upper). (Lower) Overlay of the RET core kinase domain wild type and S904F mutant structures omitting bound waters for clarity

after lentiviral transduction of *CCDC6-RET* cDNAs with or without the S904F mutation (Supplementary Fig. 5). Consistently, Ba/F3 cells expressing *CCDC6-RET* cDNA with the S904F mutation showed a greater GI$_{50}$ to vandetanib (1070 vs. 278.2 nM) than those without the mutation (Fig. 2b). To validate the results of the cell-based assay, an in vitro kinase assay was performed using purified RET kinase polypeptides, which showed that the IC$_{50}$ of vandetanib was higher in the S904F mutant (908.5 nM) than in the wild-type kinase (67.82 nM) (Fig. 2c). This result was consistent with a significantly poorer inhibitory constant ($K_i$) of vandetanib in the S904F mutant (43.09 nM) than in the wild type (11.57 nM) (Fig. 3a and Supplementary Fig. 6).

An in vitro kinase assay was performed to examine the effect of the S904F mutation on the kinetic profile of RET kinase toward synthetic peptide substrates. The results showed that the S904F mutant had a higher affinity for ATP ($K_{m[ATP]}$), resulting in an increased catalytic efficiency ($k_{cat}/K_{m[ATP]}$) (Fig. 3a, b). An even more dramatic difference was observed for the S904F mutant,

which showed accelerated and enhanced autophosphorylation kinetics compared with those of the wild-type protein (Fig. 3c). These results indicate an activating effect of the *RET*-S904F mutation, which is consistent with a previous report indicating that the same mutation is responsible for familial thyroid cancer[12]. The positive effect of the S904F mutant on the affinity for ATP and autophosphorylation of RET kinase may induce vandetanib resistance by decreasing the relative competitiveness of the type I inhibitor, vandetanib, against ATP (Supplementary Fig. 6).

To explore the impact of S904F on the overall RET conformation, the crystal structure of auto-phosphorylated RET kinase domain was determined at 2.3 Å and refined to a working *R*-value of 19.8% (*R*-free 23.7%, Supplementary Table 1), with good model geometry. The overall structure is very similar to the wild type (PDB code 2IVT). Residues 820–825 are ordered first time compared with previously published RET kinase domain structures[13]. The nucleotide pocket is occupied by an adenosine

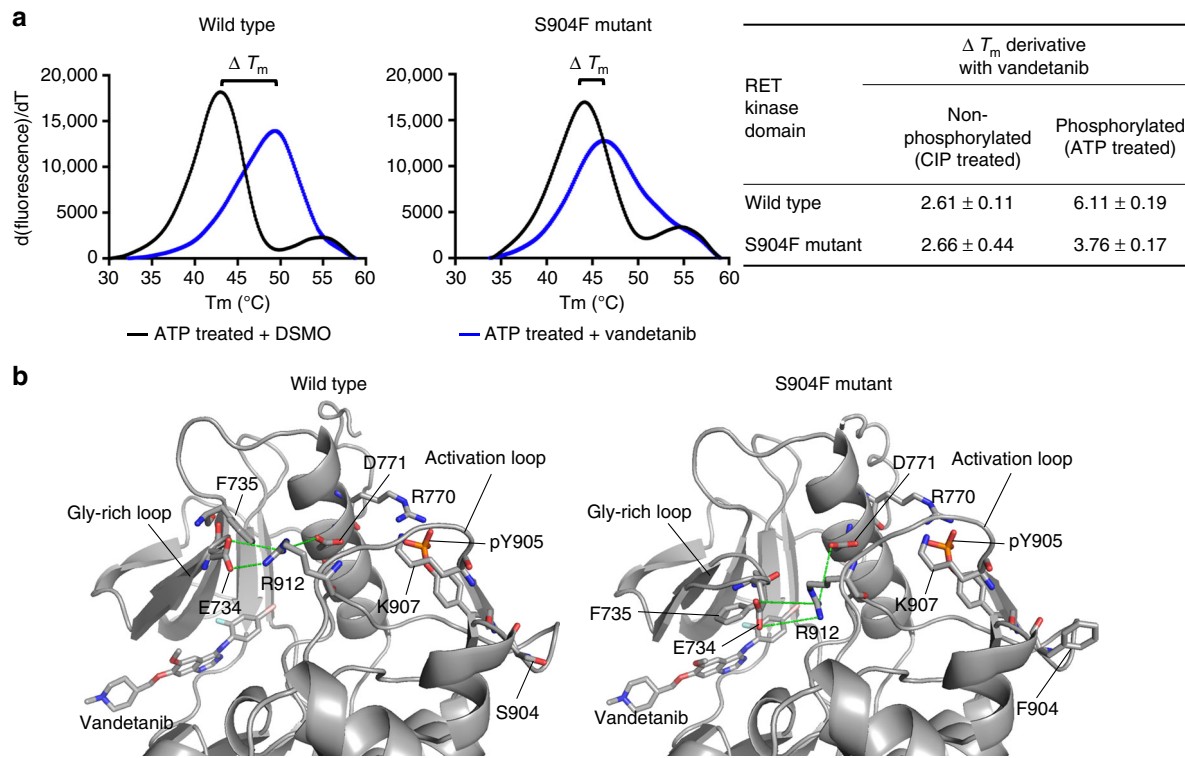

**Fig. 4** Decreased thermal stability of the RET kinase-vandetanib complex induced by the S904F mutation. **a** A thermal shift assay was performed to determine the drug-induced changes in the melting temperature ($\Delta T_m$) of purified RET KD, which reflect the stability of the complex[13]. Recombinant wild type or S904F mutant RET KD was generated using previously published methods[13]. Each protein was dephosphorylated using CIP-phosphatase and then either used directly (unphosphorylated) or phosphorylated by addition of Mg-ATP, followed by incubation with DMSO or 1 μM vandetanib. Wild type and S904F mutant RET KDs without drug showed $T_m$ values of 43.00 ± 0.06 °C and 44.16 ± 0.04 °C, respectively (Supplementary Table 1). Addition of vandetanib increased the $\Delta T_m$ of wild-type RET KD by 6.11 ± 0.19 °C, whereas it increased the $\Delta T_m$ of phosphorylated S904F RET KD only by 3.76 ± 0.17 °C. Unphosphorylated RET KDs showed little or no ($\Delta T_m$) increase irrespective of the mutation status (Supplementary Table 1). **b** Geometry of the hydrogen-bond network consisting of E734, D771, and R912, which regulates the accessibility of ATP to the nucleotide-binding pocket and active site. The mean structures of E734, F735, R770, D771, S/F904, pY905, K907, R912, and vandetanib, generated by molecular dynamics simulations of 1 μs × 3 times, are represented by thick sticks (gray, carbon; blue, nitrogen; orange, phosphorus; red, oxygen; light blue, fluorine; and light pink, bromine). Mean geometry of the hydrogen-bond network formed by E734, D771, and R912 is depicted by dashed green lines, showing the formation of an E734-R912 hydrogen bond at an irregular position in the S904F mutant, and the intermediate conformer is likely to be stabilized by this aberrant hydrogen-bond network

moiety. Only small differences in side chain positioning are observed close to the F904 residue. F904 side chain makes Van der Waals contacts with V906, I927 and the aliphatic portion of E902, thereby defining a small hydrophobic core absent from wild type RET (Fig. 3d). This may tether the activation loop more tightly in both non-phospho- and phospho-RET forms, enhancing the basal level of RET activity.

**Decreased thermal stability of the RET kinase-drug complex induced by the S904F mutation**. The $K_i$ and IC$_{50}$ values for vandetanib were increased in the S904F mutant at concentrations several 1000-fold lower than those of ATP (Figs. 2c and 3a). To further investigate this effect, a thermal shift assay was performed to examine directly the stability of the enzyme-ligand complex. This assay determines the drug-induced increase in the melting temperature ($\Delta T_m$) of the purified RET KD, which reflects the stability of the kinase-ligand complex[13].

When the RET protein was phosphorylated, addition of vandetanib increased the $\Delta T_m$ of the wild-type RET KD by 6.11 ± 0.19 °C, consistent with a recent study[14]. Addition of vandetanib increased the $\Delta T_m$ of the S904F RET KD only by 3.76 ± 0.17 °C (Fig. 4a and Supplementary Table 2). When the RET kinase protein was not phosphorylated, the $\Delta T_m$ increases were

smaller and comparable between the wild type and S904F RET KDs (Fig. 4a and Supplementary Table 2). These data suggested that the S904F mutation decreased the affinity for vandetanib in the KD by reducing the magnitude of thermal stabilization induced by drug addiction.

## Discussion

The present study identified a secondary S904F mutation conferring resistance to vandetanib in a metastatic LADC harboring a *CCDC6-RET* fusion that initially exhibited a response to treatment but later progressed. The RET-S904F mutation was located in the AL, in which mutations are not linked to drug resistance to the best of our knowledge. The mutant showed increased ATP affinity and autophosphorylation activity. These results are consistent with the S904F mutation as a germline oncogenic mutation responsible for the development of familial thyroid cancer and with the ability of the *RET* S904F mutant to activate RET kinase and transform NIH3T3 fibroblasts[12]. Since vandetanib is a type I inhibitor that inhibits RET kinase activity in an ATP-competitive manner, the increased ATP affinity and autophosphorylation activity may be responsible for the resistance induced by the mutant. Increased ATP affinity underlies the resistance to type I TKIs of an EGFR kinase mutant with a

T790M mutation in its ATP-binding cleft[15,16]. Thus, increased ATP affinity might be a common mechanism of drug resistance in oncogenic kinases.

A thermal shift assay in the presence or absence of drug suggested that the S904F mutation is less able to bind vandetanib. The co-crystal structure of RET kinase in complex with vandetanib indicates that the mutated residue (F904) does not directly interfere with vandetanib binding because of its location (Fig. 4b)[13]. Therefore, we investigated whether an allosteric effect of the S904F mutation may result in a conformational change affecting the stability of the RET kinase-vandetanib complex. Molecular dynamics (MD) simulation using the human RET KD in complex with vandetanib[17] suggested dynamic coupling pattern differences in regions of AL and the glycine-rich loop (GRL) between the wild type and S904F mutant (Supplementary Fig. 7a), concomitantly with a decrease in the fluctuation of these regions in the S904F mutant (Supplementary Fig. 7b). The GRL of the RET KD, in particular the side chain of phenylalanine 735 (F735), has been reported to regulate the ATP-binding capacity by interconverting between two conformers, open (ATP-binding competent) and closed (ATP-binding blocked)[18] (Supplementary Fig. 7c). The mean structures of MD trajectories suggested that the S904F mutation causes a geometrical change in E734 (in GRL), D771 (in αC helix), and R912 (in AL) which compose a triad of tethered residues via salt bridge interaction[18] (Fig. 4b). The geometrical change was associated with the appearance of an additional de novo conformer represented by a higher energy state (i.e., lower affinity to vandetanib) than that of the open/closed conformers (Supplementary Fig. 7d and e). The de novo conformer (designated as "intermediate") was suggested to have a protrusion of the side chain of F735 in the GRL toward the drug-binding site, which would sterically interfere with the binding of vandetanib (Supplementary Fig. 7c, f and g; and Supplementary Movies 1, 2).

The present results indicated that the secondary S904F mutation located in the AL, and therefore distant from the ATP-binding site, may exhibit allosteric effects conferring resistance to vandetanib. The increased ATP affinity and autophosphorylation activity of the mutant were considered the main factors underlying the resistance to the type I inhibitor vandetanib. Loss-of-tight vandetanib interaction could be inferred from the smaller thermal melt gains upon drug addition to S904F mutant protein. The precise mechanisms underlying these allosteric effects remain largely unclear requiring further clarification by crystallography and/or biophysical studies.

## Methods

**Study oversight**. The study was approved by the Institutional Review Boards of the National Cancer Center. The patient provided written informed consent for the genetic analysis and participation in the investigator-initiated clinical trial.

**Clinical trial**. The LURET (Lung Cancer with RET Rearrangement Study; clinical trial registration number: UMIN000009095, https://upload.umin.ac.jp/) trial investigated the efficacy of vandetanib in non-small cell lung cancer (NSCLC) with an oncogenic RET fusion. LURET is coupled with a nation-wide genetic screen, LC-SCRUM-Japan (Lung Cancer Genomic Sreening Project for Individualized Medicine in Japan), in which more than 2000 EGFR mutation-negative non-squamous NSCLC patients were enrolled from >190 hospitals in all 47 prefectures of Japan for screening of ALK, RET, and ROS1 gene fusions.

For the LURET clinical trial, eligible patients had pathologically confirmed, locally advanced or metastatic non-squamous NSCLC with RET rearrangement without EGFR mutations and failed at least one previous chemotherapy. RET fusion positivity was determined by double positivity in reverse transcriptase-polymerase chain reaction (RT-PCR) and break-apart fluorescence in situ hybridization (FISH) assays. Of 34 RET fusion-positive patients detected by RT-PCR, 19 were enrolled in the clinical trial to study the efficacy of vandetanib. In 17 eligible patients included in the primary analysis, the response rate was 53%. In the 19 registered patients, the disease control rate was 90%. Progression-free survival was 4.7 months. The details of the clinical trial are described in the published paper[8].

**Samples**. A patient with lung adenocarcinoma (LADC) harboring a CCDC6-RET fusion underwent biopsy of pretreated and vandetanib-resistant tumors. Histopathological and immunohistochemical diagnoses were performed. Total RNA was extracted from grossly dissected, snap-frozen tissue samples using TRIzol (Invitrogen, Carlsbad, CA, USA), and its quality was determined using a 2100 Bioanalyzer (Agilent Technologies, Santa Clara, CA, USA). Genomic DNA was extracted from the same tumors and peripheral blood using the QIAamp DNA mini kit (Qiagen, Limburg, The Netherlands).

**PCR and sanger sequencing**. Methods for RT-PCR and Sanger sequencing were described previously[6]. Total RNA (500 ng) was reverse-transcribed into cDNA using Superscript III Reverse Transcriptase (Invitrogen). cDNA (corresponding to 10 ng of total RNA) or 10 ng of genomic DNA was subjected to PCR amplification using KAPA Taq DNA Polymerase (KAPA Biosystems, Woburn, MA, USA). For detection of the fusion point, the following PCR primers were used: for cDNA amplification, 5′-TGCAGCAAGAGAACAAGGTG-3′ (CCDC6 forward) and 5′-TGAGAGGCCGTCGTCATAAA-3′ (RET reverse); for genomic DNA amplification, 5′-TGTCAAAACTGGCCTCTCTG-3′ (CCDC6 forward) and 5′-GGAACC-CACAGTCAAGGTCA-3′ (RET reverse). The reactions were carried out in a thermal cycler under the following conditions: 40 cycles of 95 °C for 30 s, 60 °C for 30 s, and 72 °C for 2 min, with a final extension at 72 °C for 10 min. The gene encoding glyceraldehyde-3-phosphate dehydrogenase (GAPDH) was amplified to estimate the efficiency of cDNA synthesis in the RT-PCR analysis. PCR products were directly sequenced in both directions on an ABI 3130xl DNA Sequencer (Applied Biosystems, Foster City, CA, USA) using the BigDye Terminator kit.

**Targeted and whole-exome sequencing**. Targeted and whole-exome sequencing was performed using 1.0 μg of DNA extracted from biopsied tissues. Targeted genome and exome capture were performed using the Agilent SureSelect kits, NCC Oncopanel (Catalog No. 931196, Agilent) and Human All Exon V5, respectively. Sequencing was performed on the Illumina HiSeq 1500 platform using 100 bp paired-end reads (Illumina). Basic alignment and sequence quality control were performed using the Picard and Firehose pipelines. The reads were aligned against the reference human genome from UCSC human genome 19 (Hg19) using the Burrows Wheeler Aligner Multi-Vision software package. Because duplicate reads were generated during the PCR amplification process, paired-end reads that aligned to the same genomic positions were removed using SAMtools. Somatic SNVs were called by the MuTect program, which applies a Bayesian classifier to allow detection of somatic mutations with low-allele frequency. Somatic insertion/deletion mutations (indels) were called using the GATK Somatic IndelDetector (https://software.broadinstitute.org/gatk/). In addition, target sequencing was performed for 10 ng of DNAs using the Ion Ampliseq Cancer Hotspot Panel v2 and the Ion Proton sequencer (Thermo Fisher Scientific, Waltham, MA, USA).

**Alignment of amino acid sequences**. Tyrosine kinases belonging to the RET superfamily were listed via the Ensembl Genome Browser (www.ensembl.org) and the human kinome[19]. The alignments were obtained from Uniprot (http://www.uniprot.org/) and visualized by Jalview (http://www.jalview.org/).

**Immunohistochemistry**. Sections (4 μm thick) were deparaffinized. Immunohistochemical staining was performed using a Ventana Discovery XT instrument supplied by Ventana Medical Systems Inc. (Tucson, AZ, USA) according to the manufacturer's instructions. The primary antibodies against TTF-1 (SP141) were purchased from Ventana Medical Systems Inc. (Catalog No. 790-4756, ready to use without dilution), and thyroglobulin was purchased from Dako (Catalog No. M078101, Glostrup, Denmark, ready to use without dilution). The reactions were visualized with 3,3′-diaminobenzidine followed by counterstaining with hematoxylin.

**Cell lines and reagents**. NCI-H1299 cells and Ba/F3 cells were provided by Dr. J.D. Minna of UT Southwestern Medical Center and Dr. Hiroyuki Mano of University of Tokyo, respectively. 293FT cells and WEHI-3B cells were obtained from Invitrogen (Carlsbad, CA, USA) and RIKEN BioResource Center (Japan), respectively. NCI-H1299 and WEHI-3B were cultured in RPMI medium with 10% fetal bovine serum (FBS). Ba/F3 cells were cultured in RPMI medium containing 10% FBS and 10% WEHI-3B–conditioned medium (a source of interleukin 3 [IL3]), and 293FT cells were cultured in DMEM with 10% FBS. All cells were incubated at 37 °C in 5% $CO_2$.

Vandetanib was purchased from Selleck (Houston, TX, USA). Primary antibodies against RET (Catalog No. ab134100) and phospho-Tyr1015 RET (Catalog No. ab74154) were purchased from Abcam (Cambridge, UK). Antibodies against phospho-Tyr 905 Ret (Catalog No. 3221), GAPDH (Catalog No. 5174), and beta-actin (Catalog No. 3700) were purchased from Cell Signaling Technology (Danvers, MA, USA). Antibodies against phospho-Tyr1062 RET (Catalog No. sc-20252-R) and total phospho-tyrosine (Catalog No. sc-7020) were purchased from Santa Cruz Biotechnology (Santa Cruz, CA, USA).

**Construction of lentiviral vectors expressing wild type and S904F mutant CCDC6-RET cDNA**. Full-length wild type and S904F mutant *CCDC6-RET* cDNAs were synthesized by FASMAC (Kanagawa, Japan). cDNAs were ligated into pLenti-6/V5-DEST plasmids (Invitrogen). The integrity of each inserted cDNA was verified by Sanger sequencing. Expression of cDNA products was confirmed by immunoblotting of transiently transfected cells.

**Lentiviral production and infection**. Lentiviruses were generated in 293FT cells ($6 \times 10^6$ cells per 10 cm plate) transfected with pLenti-6/V5-DEST plasmid containing either the wild type or S904F mutant *CCDC6-RET* cDNA and ViraPower packaging mix (Invitrogen) using the Lipofectamine 3000 reagent (Invitrogen). Viral supernatants were collected at 42 h after the medium change, and then used to infect $4.0 \times 10^5$ Ba/F3 cells in the presence of 10 µg/ml Polybrene (Sigma-Aldrich, St Louis, MO, USA) by centrifugation at $3000 \times g$ for 150 min at 32 °C. Following overnight incubation at 37 °C in 5% $CO_2$, the cells were distributed into 24-well plates and selected in medium containing IL3 and 8 µg/ml blasticidin (Invitrogen) for 1 week. The blasticidin-resistant cells were grown in IL3-free medium for 2 weeks. The expression of exogenous CCDC6-RET proteins was confirmed by immunoblotting coupled with Sanger sequencing of RT-PCR products from the cells (Supplementary Fig. 5a, b).

**CCDC6-RET phosphorylation assay**. NCI-H1299 cells were transiently transfected with 5 µg of plasmid DNA using Lipofectamine 3000. The cells were re-seeded at 12 h after transfection. After 24 h, the cells were cultured with vehicle or increasing doses of inhibitors for 6 h. Ba/F3 cells ($5.0 \times 10^6$) stably expressing wild type or S904F mutant CCDC6-RET were cultured with vehicle or increasing doses of inhibitors for 6 h. The cells were lysed in RIPA buffer (1% NP-40, 0.1% sodium deoxycholate, 50 mM Tris-HCl [pH 7.6], 150 mM NaCl, 1 mM EDTA [pH 8.0], 0.1% SDS, 1 mM $Na_3VO_4$, and 10 mM NaF) containing Complete Protease Inhibitor Cocktail (Roche, Mannheim, Germany). Cell lysates were centrifuged at 14,000×r.p.m. for 15 min, and the supernatants were collected. The supernatants were subjected to SDS-PAGE, followed by immunoblotting on polyvinylidene difluoride membranes. The membranes were blocked for 1 h with TBS containing 0.1% Tween 20 (TBST) and 1.0% BSA, and then probed with primary antibodies; anti-RET (used at 2000-fold dilution), anti-phospho-Tyr1015 RET (used at 500-fold dilution), anti-phospho-Tyr 905 Ret (used at 1000-fold dilution), anti-GAPDH (used at 1000-fold dilution), and anti-beta-actin (used at 1000-fold dilution). After washing with TBST, the membranes were incubated with horseradish peroxidase-conjugated anti-mouse or anti-rabbit secondary antibodies, and then visualized with enhanced chemiluminescence reagent (Perkin Elmer, Waltham, MA, USA). Intensities of signals were quantified using a LAS3000 imaging system (Quansys Biosciences, West Logan, UT, USA). Assays were independently performed more than three times. To determine the half-maximal inhibitory concentration ($IC_{50}$) values in the H1299 model, the signal intensities of total and phospho-Y905 of CCDC6-RET were quantified using Multi-gauge software (Fujifilm, Tokyo, Japan). After subtraction of the background, the signal intensity at each dose was standardized by dividing it by that of the inhibitor-free sample labeled "0". The mean ratios of phospho-Y905 to total CCDC6-RET in three independent assays were plotted with error bars. Uncropped gel data are supplied in the Supplementary Fig. 8.

**Cell viability assays**. Twenty-four hours before inhibitor treatment, 2000 Ba/F3 cells stably expressing CCDC6-RET with or without the S904F mutation were plated in quadruplicate in 96-well plates. Serially diluted inhibitors were added to the wells. Cell viability was measured at 72 h after drug treatment using the CellTiter-Glo luminescent cell viability reagent (Promega, Madison, WI, USA) with EnVision (Perkin Elmer, Waltham, MA, USA). Cell viability was calculated as the cell count in drug-treated samples relative to that in untreated samples. The data were displayed graphically using GraphPad Prism version 6.0 (GraphPad Software Inc., San Diego, CA, USA). Assays were independently repeated more than three times.

**In vitro kinase assay**. The recombinant RET kinase domain (KD; amino acids 658–1072) with or without the S904F mutation was expressed by baculovirus in Sf9 insect cells using an N-terminal GST tag (gene accession number: NM_020630). To determine kinetic constants, RET kinase assays were performed in triplicate at 25 °C for different incubation times (0, 10, 20, 30, 40, and 50 min) in a final volume of 25 µl as follows: 5 µl of diluted active wild type or S904F mutant RET kinase (41.6 ng each); 5 µl of kinase assay buffer (SignalChem); 5 µl of diluted vandetanib (various concentrations); 5 µl of IGF1Rtide synthetic peptide substrate (KKKSPGEYVNIEFG) (SignalChem, Richmond, BC, Canada); and 5 µl of radio-active $^{32}$P-ATP cocktail at various concentrations ($^{32}$P-ATP [Perkin Elmer, Waltham, MA, USA] and cold ATP [SignalChem, Richmond, BC, Canada]). To determine $IC_{50}$ values, assays were performed in triplicate for 30 min with 5 µM ATP. The assay was initiated by addition of $^{32}$P-ATP to the reaction mixture, including kinase assay buffer, serially diluted vandetanib, and IGF1Rtide synthetic peptide as substrate, and incubated at 30 °C for 20 min. After the incubation period, the reaction was terminated by spotting 10 µl of the reaction mixture onto a multiscreen phosphocellulose P81 plate. The multiscreen phosphocellulose P81

plate was washed twice for ~15 min each in 1% phosphoric acid solution. The radioactivity on the P81 plate was counted in the presence of scintillation fluid in a TriLux scintillation counter. For each target, blank controls included all assay components except the corresponding substrate, which was replaced with an equal volume of assay dilution buffer. The corrected activity for each target was determined by subtracting the blank control value. Reproducibility was confirmed by performing the same experiment three times. The data were analyzed using GraphPad Prism version 6.0 for Mac to calculate kinetic parameters, inhibitory constant ($K_i$), and $IC_{50}$ values.

**RET kinase domain S904F mutant X-ray structure determination**. Recombinant RET kinase domain RET-KD (residues 705–1013) was expressed in Sf9 cells using a recombinant baculovirus and purified by affinity chromaography as described previously[20]. Residues from the kinase insert region (827–840) were omitted from this construct. Purified RET mutant protein was then concentrated to 4.5 mg/ml in crystallization buffer (20 mM Tris, pH 8, 100 mM NaCl, 1 mM DTT, and 1 mM EDTA). Crystals were grown at 16 °C in sitting drops containing 3.4 M sodium formate, and 0.1 M sodium acetate pH 4.4. All crystals were collected directly into the cryoprotectant oil, perfluoro-polyether (Hampton Research) and flash-frozen in liquid nitrogen. The data were collected, at the Swiss Light Source (Supplementary Table 1) and processed using standard data integration and scaling software. The crystals belong to space group $P4_32_12$ with a single molecule in the asymmetric unit. The structure was solved by molecular replacement using phosphorylated RET-KD-P protein (PDB code 2IVT) as a search model and omitting flexible regions and phosphotyrosines. The structure was refined using Phenix.refine[21] and rebuilt using Coot[22]. Electron density is of a good quality. The RET activation loop is phosphorylated on Tyr 905 sidechains. The structure was refined at 2.3 Å to an $R_{work}$ of 19.8% (and $R_{free}$ 23.7%) shown in Supplementary Table 1.

**Thermal shift assay**. Wild type and S904F mutant RET KD proteins were expressed in SF21 cells and purified using a GST affinity tag as previously described[13]. Each protein was dephosphorylated and then either left in its dephosphorylated state or phosphorylated by addition of ATP/Mg for 90 min at room temperature. To determine the thermal shifts, recombinant proteins were incubated with DMSO or 1 µM vandetanib. Sypro-Orange dye (Life Technologies) was added to each drug treatment, and the thermal shift was measured in a 7500 Fast RT-PCR machine (Applied Biosystems) in a temperature range of 25–90 °C. Subsequent analysis was performed using Protein Thermal Shift Software v1.2 (Applied Biosystems).

**Autophosphorylation assay**. The time course of autophosphorylation of recombinant purified RET KD (2.5 µM) was examined in the presence of saturating concentrations of ATP (5 mM) and $MgCl_2$ (10 mM) for 0–80 min as previously described[18]. Reactions were stopped by addition of 4× loading sample buffer (Invitrogen) with 10% β-mercaptoethanol and boiling for 5 min. Samples were then loaded onto a NuPAGE Invitrogen 4–12% Bis-Tris precast gel. Phosphorylation was detected with the following antibodies: anti-phospho-Tyr1062 RET (used at 3000-fold dilution), anti-phospho- tyrosine (used at 1000-fold dilution), phospho-Tyr 905 RET (used at 1000-fold dilution), and anti-total RET (used at 1000-fold dilution), purchased from Cell Signaling Technology. Uncropped gel data are supplied in Supplementary Fig. 9.

**Molecular dynamics (MD) simulation**. The initial structural data of vandetanib-bound RET kinase were obtained from the Protein Data Bank (PDB code: 2IVU). The structures of disordered loops (residues Leu712–Asp714 and Arg820–Arg844) were modeled using the Structure Preparation module in the Molecular Operating Environment (MOE, Chemical Computing Group, Montreal, Canada), version 2013.08[23]. The initial structural data of ATP-bound RET were obtained from the PDB (PDB code: 2IVT). The structures of disordered loops (residues Ile711–Pro715 and Val822–Arg844) were modeled using the Structure Preparation module in MOE, and β and γ phosphates were modeled using the Builder module in MOE. The N- and C-termini of all protein models were capped with acetyl and N-methyl groups, respectively. Titratable residues remained in their dominant protonation state at pH 7.0. A S904F mutation was introduced into the structure of wild-type RET using the Structure Preparation module in MOE.

All MD simulations were performed using the GROMACS 4 program[24] on High Performance Computing Infrastructure equipped with NVIDIA Tesla K20 GPGPUs. Both small-molecule compounds (vandetanib and ATP) were optimized, and the electrostatic potential was calculated at the HF/6–31G* level using the GAMESS program[25], after which the atomic partial charges were obtained by the RESP approach[26]. Other parameters for vandetanib and ATP were determined by the general Amber force field (GAFF)[27] using the antechamber module of AMBER Tools 12. The parameters for ATP were as determined by Meagher et al.[28]. The Amber ff99SB-ILDN force field was used for proteins and ions[29], and TIP3P was used for water molecules[30]. Water molecules were placed around the complex model with an encompassing distance of 8 Å to form a $88 \times 83 \times 72$ Å$^3$ periodic box for the RET-vandetanib complex and a $91 \times 86 \times 74$ Å$^3$ periodic box for the RET-ATP complex, including ~16,000 water molecules in the RET-vandetanib system

and 17,000 water molecules in the RET-ATP system. Charge-neutralizing ions were added to neutralize the system. Electrostatic interactions were calculated using the particle mesh Ewald (PME) method[31] with a cutoff radius of 10 Å. Van der Waals interactions were cutoff at 10 Å. The P-LINCS algorithm was applied to constrain all bond lengths[32]. After energy-minimization of each of the fully solvated systems, the system was equilibrated for 100 ps at 298 K under NVT conditions and run for 100 ps under NPT conditions at 1 bar, with the heavy atoms of the protein and compound held in fixed positions. The time constants for the temperature and pressure couplings to the bath were 0.1 ps and 2 ps, respectively. Each production run lasted 50 ns at 298 K, maintained using velocity rescaling with a stochastic term[33], and 1 bar, maintained using the Parrinello–Rahman pressure coupling[34]. Equilibration and production runs were performed with time steps of 2 fs.

For the RET-ATP system, three independent MD simulations were performed with different initial velocities (1 μs × 3), resulting in 3 μs simulations. For the RET-vandetanib system, 15 independent MD simulations were performed with different initial velocities (50 ns × 15), and an additional 950 ns simulation was performed for each of three trajectories in which open, closed, and intermediate conformational states were observed. These simulations were conducted for both wild type and mutant RET. Snapshots were output every 2 ps to yield 500 snapshots per ns of simulation.

**Clustering of MD structures of the RET-vandetanib complex.** Conformational fluctuations of the ATP-binding pocket in the RET-vandetanib complex were analyzed by focusing on the glycine-rich loop (GRL; 731–737) because its crystallographic B-factors were higher than those of other regions surrounding the pocket[13]. In addition, the presence of two distinct conformations of the loop in the crystal structure[18] indicate that it is highly flexible. The conformational states of the loop and compound were analyzed using intermolecular potential energy terms[35]. At even numbers of nanoseconds in the trajectories (e.g., 0, 2, and 4 ns), 2 ns averages of intermolecular electrostatic and van der Waals potential energies were calculated for all pairs of compound atoms and amino acids in the GRL (30 compound atoms × 7 residues × 2 = 420 energy terms), and each of the CH₃, CH₂, CH, and NH groups was treated as a single atom. These energy terms in 50 ns × 15 trajectories for RET-vandetanib (375 frames) were hierarchically clustered using the furthest-neighbor method and Euclidian distance, and the trees produced by the clustering were cut at a height of 0.5 kcal/mol. Almost all frames for the wild-type-vandetanib complex were clustered into a single category, whereas frames for the S904F mutant-vandetanib complex were clustered into two categories. The same method was applied to long time trajectories (1 μs × 3, 1500 frames) for RET-vandetanib/ATP.

**Calculation of the protein-compound binding free energy.** The MP-CAFEE method was used to calculate the protein-compound binding free energy[17,36,37]. For each protein-compound complex and solvated compound, a 32 λ parameter set for the Coulomb and van der Waals interactions was used[17], and six independent simulations were performed with different initial velocities for each λ parameter. The binding free energy for a single-protein-compound pair was calculated by performing 384 (= 6 × 32 × 2) simulations. During these simulations, the temperature was maintained at 298 K using the Nose-Hoover thermostat, and the pressure was maintained at 1 bar using the Berendsen barostat[38]. The time constants for the temperature and pressure couplings to the bath were 0.3 ps and 1 ps, respectively. MD simulations for MP-CAFEE were performed on the K computer (RIKEN, Japan).

**Data availability.** The authors declare the data supporting the findings of this study are available within the article and its Supplementary Information files. Uncropped scans of immunoblots are shown in Supplementary Figs. 8, 9. The cDNA sequence for the S904F mutant is available in GenBank under the accession code KU254649. Raw data for targeted and whole-exome sequencing are available in Integrative Disease Omics Database (https://gemdbj.ncc.go.jp/omics/docs/others.html) under accession codes TRS001/002 and WES001, respectively. The S904F structure coordinates and structures factors have been deposited with Protein Data Bank (PDB) under the accession code 6FEK. All other data are available from the corresponding author on request.

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

## Acknowledgements

We thank Dr. Mamoru Kato, Dr. Kuniko Sunami, Ms. Ayaka Otsuka, and Ms. Sachiyo Mitani for providing technical and methodological assistance. This work was supported in part by grants-in-aid from the Japan Agency for Medical Research and Development (AMED) (JP17ck0106255, JP17ck0106148, and JP17ak0101067) and the National Cancer Center Research and Development Fund (26A-1: NCC Biobank). The simulation study was supported by the FOCUS Establishing Supercomputing Center of Excellence, JST, CREST "Big data application" and MEXT, as "Priority Issue 1 on Post-K computer" (Building Innovative Drug Discovery Infrastructure through Functional Control of Biomolecular Systems). This research used computational resources of the K computer provided by the RIKEN Advanced Institute for Computational Science through the High Performance Computing Infrastructure System Research Project (Project ID: hp150272 and hp160213). N.Q.M. acknowledges that this work was supported by the Francis Crick Institute, which receives its core funding from Cancer Research UK (FC001115), the UK Medical Research Council (FC001115) and the Wellcome Trust (FC001115); by the NCI/NIH (grant reference 5R01CA197178); by the Association for Multiple Endocrine Neoplasia Disorders MTC Research Fund. We acknowledge expert assistance from Andrew Purkiss in the S904F mutant X-ray data collection.

## Author contributions

T.N., T.K., N.Q.M., and K.G. designed the study. S.Ni., S.Ma., G.I., K.Y., and K.G. enrolled patients and provided specimens. K.Tsuc., S.Ma., Y.S., S.Mi., and H.I. performed DNA and cDNA sequencing. T.N., Y.S., N.Q.M., R.C., P.P.K., S.Na., and K.Tsum. performed molecular biological/biochemical experiments. M.A. and Y.O. performed computational simulation. All authors reviewed the final manuscript.

## Additional information

**Competing interests:** The authors declare no competing financial interests.

