## [Peer Review File · Nature Communications]

Reviewers' comments:

Reviewer #1 (Remarks to the Author):

The paper by Nakaoku et al reports a novel mutation in RET kinase (S904F) that results in resistance to the kinase inhibitor vandetanib in lung adenocarcinoma harboring a CCDC6-RET. The mutation is located in the activation segment, thus quite distantly located to the ATP binding site. ATP affinity seems to be slightly decreased (K_m 46 \pm 3 (wild type) to 30.4 \pm 3 mutant) while autophosphorylation is drastically increase. In a Ba/F3 cell model the mutation decreases significantly sensitivity to vandetanib. Due to the distant location the authors suggest an allosteric model of kinase activation by the activation loop model. Evidence for this hypothesis is mainly based on in silico simulations and temperature shift assays indicating stabilization of the mutant when compared to wild type protein.

All in all this new resistance mutation is interesting and may have therapeutic relevance. The proposed mechanisms is however only based on in silico data, which is clearly insufficient to claim a structural mechanism leading to vandetanib resistance.

The structural model that the authors propose changes in the glycine binding loop, a highly mobile region in kinases that can adopt many different conformations. I am therefore doubtful that conformational changes in the glycine rich loop that are presumably induced by the mutant will influence inhibitor binding. The T_m assay data are suitable to support the proposed mechanism. T_m data can only serve to rank relative affinities of ligands for the same enzyme. Comparison of melting temperatures between a mutant and wild type have no physical meaning and should not be used to validate a structural mechanism proposed by MD simulation. 1-2 degrees in melting temperature is usually within the error range between biological (but not technical) replicates. In addition, the melting curves seem biphasic suggesting a more complex unfolding system. In figure 2e the ATP treated sample (I suggest not naming the wild type protein "non-mutant") shows two transitions. In the ATP and vandetanib sample the transition is broadened, probably indicating a complex melting behaviour making T_m determination uncertain.

IN summary, the mutation detected in the activation segment is very interesting and data showing an increase in ATP K_m and resistance in BaF cell models in intriguing. The authors should therefore use these results as a basis to provide convincing mechanistic data rather than speculating on structural changes that are not supported by experimental evidence.

For instance, since the activation loop is involved in substrate recognition the increase auto-activation activity might be due to more efficient substrate recognition rather than allosteric regulation to a very distant ATP site.

We revised our manuscript according to the reviewer's comments. In the revised manuscript, we focused on clinical experimental data and used *in silico* data in the discussion to speculate on the characteristics of the mutant. Accordingly, we changed the title of our paper to "**A Secondary *RET* Mutation in the Activation Loop Conferring Resistance to Vandetanib**" to more accurately represent the results and conclusion.

The revised points are summarized as follows:

1. The results section describes clinical and experimental data.
We have now included an experimentally determined crystal structure for the S904F mutant (Figure 2g and Supplementary Table 1).
2. All figures, except Fig. 3b, show the experimental results.
3. MD simulation data are presented in a supplemental figure (Sup. Fig. 7) and discussed in a single paragraph in the discussion section. The description of the MD simulation was toned down throughout the manuscript.

As pointed out by the reviewer, the changes in melting temperature detected in the thermal shift assay were not substantial. However, the differences were highly reproducible, as indicated in a recently published paper, in which the co-authors of the present study are included (Table S4 in Plenker D et al. *Sci Transl Med*, 14; 9 (394), 2017). This indicates a significant difference between the mutant and wild-type proteins. The reviewer also points out that the biphasic melting curves in ATP-treated samples indicate a complex unfolding system. However, the melting curves of the vandetanib-RET complex (blue lines in Fig. 3a) are monophasic, suggesting that the unfolding system is simple at least in the presence of vandetanib. Therefore, we would like to retain these data as Fig. 3a as evidence of one of the characteristics of the mutant.

Reviewer's comments:

Reviewer #1 (Remarks to the Author):

The paper by Nakaoku et al reports a novel mutation in RET kinase (S904F) that results in resistance to the kinase inhibitor vandetanib in lung adenocarcinoma harboring a CCDC6-RET. The mutation is located in the activation segment, thus quite distantly located to the ATP binding site. ATP affinity seems to be slightly decreased (K_m 46 \pm 3 (wild type) to 30.4 \pm 3 mutant) while autophosphorylation is drastically increase. In a Ba/F3 cell model the mutation decreases significantly sensitivity to vandetanib. Due to the distant location the authors suggest an allosteric model of kinase activation by the activation loop model. Evidence for this hypothesis is mainly based on in silico simulations and temperature shift assays indicating stabilization of the mutant when compared to wild type protein.

All in all this new resistance mutation is interesting and may have therapeutic relevance. The proposed mechanisms is however only based on in silico data, which is clearly insufficient to claim a structural mechanism leading to vandetanib resistance.

The structural model that the authors propose changes in the glycine binding loop, a highly mobile region in kinases that can adopt many different conformations. I am therefore doubtful that conformational changes in the glycine rich loop that are presumably induced by the mutant will influence inhibitor binding. The T_m assay data are suitable to support the proposed mechanism. T_m data can only serve to rank relative affinities of ligands for the same enzyme. Comparison of melting temperatures between a mutant and wild type have no physical meaning and should not be used to validate a structural mechanism proposed by MD simulation. 1-2 degrees in melting temperature is usually within the error range between biological (but not technical) replicates. In addition, the melting curves seem biphasic suggesting a more complex unfolding system. In figure 2e the ATP treated sample (I suggest not naming the wild type protein "non-mutant") shows

two transitions. In the ATP and vandetanib sample the transition is broadened, probably indicating a complex melting behaviour making T_m determination uncertain.

IN summary, the mutation detected in the activation segment is very interesting and data showing an increase in ATP K_m and resistance in BaF cell models is intriguing. The authors should therefore use these results as a basis to provide convincing mechanistic data rather than speculating on structural changes that are not supported by experimental evidence.

For instance, since the activation loop is involved in substrate recognition the increase auto-activation activity might be due to more efficient substrate recognition rather than allosteric regulation to a very distant ATP site.

Answer:

We deeply thank the reviewer for these comments. We agree with the reviewer that the conclusion should be drawn from solid data rather than from *in silico* analysis data. According to the reviewer's suggestions, the present manuscript was re-organized by describing the clinical and experimental data as the main results, whereas the *in silico* data were used only in the discussion to speculate on the characteristics of the mutant. Accordingly, we changed the title of our paper to "**A Secondary RET Mutation in the Activation Loop Conferring Resistance to Vandetanib**" to more accurately reflect the results and the conclusion.

The revised points are summarized as follows:

1. The results section describes clinical and experimental data.
We have now included an experimentally determined crystal structure for the S904F mutant (Figure 2g and Supplementary Table 1).
2. All figures, except Fig. 3b, show the experimental results.
3. MD simulation data are presented in a supplemental figure (Sup. Fig. 7), and the results are discussed in a single paragraph in the discussion section.

The first two paragraphs of the results section describe the clinical course of the

patient with resistance to vandetanib and the discovery of the secondary mutation. The data are shown in Fig. 1. The experimental data obtained by cell-based and *in vitro* kinase assays, which demonstrate the drug-resistant properties of the mutant, are described in the third and fourth paragraphs. The data are shown in Fig. 2. In addition, we have now included an experimentally determined crystal structure for the S904F mutant (Figure 2g and Supplementary Table 1) to address the character of the mutant.

In the last paragraph of this manuscript, increased ATP activity and autophosphorylation are described as the main factors underlying resistance to vandetanib, a type I inhibitor (lines 225–227).

MD simulation data are presented as a supplemental figure (Sup. Fig. 7) and discussed in a single paragraph in the discussion section. The description of the MD simulation results was toned down throughout the manuscript.

Answers to the specific issues;

“Comparison of melting temperatures between a mutant and wild type have no physical meaning and should not be used to validate a structural mechanism proposed by MD simulation. 1-2 degrees in melting temperature is usually within the error range between biological (but not technical) replicates. In addition, the melting curves seem biphasic suggesting a more complex unfolding system. In figure 2e the ATP treated sample (I suggest not naming the wild type protein “non-mutant”) shows two transitions. In the ATP and vandetanib sample the transition is broadened, probably indicating a complex melting behaviour making T_m determination uncertain. “

As pointed out by the reviewer, the relative changes in melting temperature (ΔT_m) detected in the thermal shift assay were not substantial. However, the differences were highly reproducible and comparable to those observed in a recently published study, in which the co-authors of the present study are included (Table S4 in Plenker D et al. *Sci Transl Med*, 14; 9 (394), 2017). This indicates a significant difference between the mutant and wild-type proteins. The reviewer also points out that the biphasic melting curves in ATP-treated

samples indicate a complex unfolding system. However, the melting curves of the vandetanib-RET complex (blue lines in Fig. 3a) are monophasic (as calculated by a first derivative data analysis), suggesting that the unfolding system is relatively simple at least in the presence of vandetanib. Therefore, we would like to retain these data as Fig. 3a as evidence of one of the characteristics of the mutant.

“The structural model that the authors propose changes in the glycine binding loop, a highly mobile region in kinases that can adopt many different conformations. I am therefore doubtful that conformational changes in the glycine rich loop that are presumably induced by the mutant will influence inhibitor binding.”

We understand that conformational changes in the glycine-rich loop constitute a candidate drug resistance mechanism among the diverse allosteric effects induced by the S904F mutation, which is distant from the ATP-binding site. In fact, in the presence of vandetanib, but not of ATP, diverse dynamic coupling among domains, including between the glycine-rich loop and activation loop, was evident in the S904F mutant (Sup. Fig. 7a). In the revised manuscript, we discussed conformational changes in the glycine-rich loop as a candidate mechanism among several dynamic changes (lines 204–222). In addition, we explained that the structural mechanisms underlying the allosteric effects remain largely unclear at present in the last paragraph (lines 229–231).

“I suggest not naming the wild type protein “non-mutant”

As suggested by the reviewer, “non-mutant” was changed to “wild type” throughout the text and figures. Thank you for your pointing out the ambiguous word.

“For instance, since the activation loop is involved in substrate recognition the increase auto-activation activity might be due to more efficient substrate

recognition rather than allosteric regulation to a very distant ATP site. "

We have now incorporated the crystal structure of the S904F mutant into the manuscript (Fig. 2g and Supplementary Table 1). The structure shows not global changes to the RET conformation but reveals a small hydrophobic core that could anchor the activation loop and could enhance auto-activation exactly as the reviewers proposed above. The structure deals more with why S904F is hyper-activated rather than defining the mechanism of drug resistance. However, it does rule out any dramatic conformation change induced by the point mutation to influence drug interaction. In our opinion this is in itself an important piece of experimental data.

We hope that the revised manuscript will satisfy the reviewer.

REVIEWERS' COMMENTS:

Reviewer #1 (Remarks to the Author):

In my initial review my main criticism was that the authors base the proposed structural model of kinase activation entirely on in silico simulations that were only supported by T_m measurements – a very indirect measurement of binding affinity. In the revised version the authors added now a crystal structure of the S904F mutant whereas in silico data are now mainly used in the discussion section. The new experimental data significantly strengthen the paper. I would be much happier if also a more direct method for inhibitor binding would have been used. However, the clinical and biochemical data are very strong and provide sufficient novelty and interest for publication. I have therefore no further issues with this manuscript.